# Numerical Analysis via Mixed Inverse Hydrodynamic Lubrication Theory of Reciprocating Rubber Seal Considering the Friction Thermal Effect

Bongjun Kim [1,†], Junho Suh [2,†], Bora Lee [2], Yondo Chun [3], Geuntae Hong [4], Jungjoon Park [5] and Yonghun Yu [5,*]

1   Hyundai Heavy Industries, Ulsan 44032, Republic of Korea
2   School of Mechanical Engineering, Pusan National University, Busan 46241, Republic of Korea
3   Electric Machines and Drives System Research Center, Korea Electrotechnology Research Institute, Changwon 51543, Republic of Korea
4   Department of Civil Engineering, Kyungpook National University, Daegu 41566, Republic of Korea
5   Innovative Transportation & Logistics Research Center, Korea Railroad Research Institute, Uiwang-si 16105, Republic of Korea
*   Correspondence: yonghunyu87@krri.re.kr
†   These authors contributed equally to this work.

**Abstract:** This study investigates how operating conditions such as ambient temperature and sealing pressure affect sealing performance for a typical U-cup seal. The developed analysis method combines inverse fluid lubrication (IHL) theory and the Greenwood–Williamson contact model (G–W model), and the effect of increasing surface temperature due to frictional heat generated between two surfaces is considered. Commercial FE software (ABAQUS) was used to simulate the interference fit analysis of rubber seals and the pressurized process. Through this model, the film distribution, working fluid leakage, and friction force in the sealing area were discussed according to the operating parameters, such as sealed pressure, rod velocity, and ambient temperature. The simulation results demonstrate the effect of fluid viscosity on oil film formation (which varies with ambient temperature), the effect of increasing the surface temperature, and the effect of surface roughness at a very small film thickness.

**Keywords:** friction; inverse hydrodynamic lubrication; leakage; mixed lubrication; reciprocating seal





## 1. Introduction

The reciprocating rod seal comprises a rod made of a hard material and a seal made of a relatively soft material, which is embedded via plastic deformation between the rod and a housing. The seal prevents the working fluid from leaking and maintains the pressure inside the housing; it is an important component that is widely used in various industries, such as automobiles and aviation. However, even if a seal is installed, leakage of the hydraulic cylinder is mechanically unavoidable and has a significant effect on the efficiency of the system, as well as causing environmental pollution. Therefore, it is essential to accurately analyze and characterize seal behavior in hydraulic systems. The seal mainly operates in the area of mixed hydrodynamic lubrication, in which the asperities that are simultaneously present between the surfaces and fluid support the external force. However, depending on its operating conditions, the reciprocating motion seal can be operated in severe lubrication regimes as its range of use expands, such as fully hydrodynamic lubrication, where there is almost no contact between the asperities and boundary lubrication. The behavior of rubber seals operating in such a wide range of lubrication conditions has been analyzed through elastohydrodynamic lubrication (EHL) theory or inverse hydrodynamic lubrication (IHL) theory.

Stupkiewicz et al. [1] predicted the dynamic sealing performance of O-ring and rectangular seals through EHL analysis, including friction due to fluid shear stress. Nikas

et al. presented an EHL model that considered many temperature- and pressure-dependent parameters across a broad range of temperatures and sealed pressures [2] and verified it through experimental comparison [3]. Researchers have continued to analyze the behavior of rubber seals operating under mixed lubrication conditions. Using a mixed EHL model, Salant et al. analyzed the behavior of single-lip [4] and double-lip U-cup seals [5]. Subsequently, they expanded the existing model to a thermal mixed EHL model that considered the surface temperature rise due to frictional heat generated in the sealing area [6] and applied it to the O-ring and U-cup seal [7]. To analyze the effect of stretching on the O-ring seal, Peng et al. [8]. simulated the EHL model, which was solved by simplifying it into an axisymmetric (2D) model by extending it in three dimensions.

General hydrodynamic lubrication (HL) analysis is mainly applied to sliding bearings, and the oil film gradient or squeeze between two hard surfaces is the main pressure-increasing mechanism, i.e., HL analysis involves calculating the oil film pressure according to the operating conditions, such as viscosity, speed, and load. However, it is impossible to analyze the behavior of rubber seals through HL analysis, such as sliding bearings, because large deformations occur even with small pressures. If EHL analysis is performed to solve this problem, the analysis becomes increasingly difficult and time-consuming. To address difficulties in analyzing the behavior of rubber seals, Kanter et al. [9]. first presented IHL theory. Whereas HL analysis calculates the pressure distribution from a specific oil film distribution, IHL analysis calculates the oil film distribution from the pressure distribution. In the IHL, it is assumed that the contact pressure obtained through the finite element (FE) analysis model is the same as the lubrication analysis pressure calculated by the oil film. Because the elastic deformation of the rubber surface is much larger than the fluid film thickness, it is hardly affected by the presence of a thin oil film [10]. According to mixed lubrication theory, the normal load acting on the two surfaces in sliding relative motion is divided and supported by the contact force of the asperities on the rough surface and the hydraulic force increased from the oil film interposed between the two surfaces; ignoring the deformation of the rubber seal due to the tangential force, it is clear that the contact force in the FE model (occurring between the two surfaces) is the sum of these two forces. Here, ignoring the contact force of a rough surface becomes IHL theory. The IHL method has the advantage that the calculation time is much faster than that of the EHL method because the deformation of the rubber seal does not need to be additionally calculated after the contact pressure of the rubber seal surface is calculated through FE analysis. Owing to its convenience, IHL analysis has been used to analyze the behavior of various seals, such as O-rings [9,11,12], rectangular seals [13–15], and U-cup seals [16,17]. However, IHL analysis that assumes perfect fluid lubrication, without considering rough surface asperity contact, was found to be very inaccurate in predicting the amount of leakage and frictional force. In particular, the frictional force was found to be significantly smaller than that of the experiment [10]. Thus, a mixed IHL study was conducted by combining the Greenwood–Williamson (GW) contact model with IHL theory to account for the effect of contact between the rough surfaces of the two plates [10,18,19].

The interface between the rubber seal and rod experiences a continuous sliding motion. In a state where a sufficiently thick oil film is not formed, such as mixed fluid lubrication and boundary lubrication, numerous asperities on each surface come into direct contact. As the asperity contact friction is much greater than the fluid friction, the friction force increases rapidly with increasing asperity contact ratio. Most of this mechanical friction energy is converted into frictional heat, which causes the temperature of the sealing zone to rise, and the lubrication state can be changed by increasing the temperature of the lubricating oil in the sealing zone. In a state where frictional heat generation is extremely serious, problems—such as breakage of the lubricating film, changes in material properties, and surface melting—may occur. Therefore, it is important to design a mechanical system with consideration for temperature.

In this study, the sealing performance was numerically analyzed by applying it to a U-cup seal. The novelty of this study is the investigation of the effect of the mixed IHL

method, considering the temperature increase due to frictional heat in the sealing zone. The static contact pressure between the seal and the rod was simulated using ABAQUS. The oil film distribution in the sealing zone was calculated using IHL theory and was modified to a mixed lubrication model by combining it with the GW model. The frictional heat generated by the fluid shear and asperity contact friction in the mixed lubrication state was calculated and the temperature rise of the sealing zone by this frictional heat was calculated via moving heat source theory. From the numerical analysis results, it was confirmed that when predicting the sealing performance of rubber seals, it is very important to consider the thermal effect under the operating conditions at high temperatures or with a high asperity contact ratio.

## 2. Mathematical Model

### 2.1. FEA Model

The FEA model was assumed to be 2D axially symmetric. Figure 1a shows the FE analysis model of the rod seal. The mesh convergence analysis was conducted to choose the 6966 mesh elements used in the simulations. The materials used for the analysis were carbon steel for the rod and seal housing and hydrogenated nitrile butadiene rubber (HNBR) for the seal, as used by Fatu et al. [17]. The carbon steel of the rod and seal housing has an elastic modulus of 210 GPa, whereas the rubber seal has a relatively small elastic modulus of 12.7 MPa; thus, all objects except rubber are assumed to be rigid bodies. For the physical properties of rubber seals, the Mooney–Rivlin model equation was used, as in Fatu et al. [17].

$$W = C_{10}(I_1 - 3) + C_{01}(I_2 - 3) + D_1(J - 1)^2 \tag{1}$$

where $W$ is a general form of the strain–energy function for incompressible materials; $I_1$ and $I_2$ are the first and second deviatoric strain invariants, respectively; $J$ is the elastic volume ratio; and $C_{10}$, $C_{01}$, and $D_1$ are material coefficients, i.e., $C_{10} = -0.167$, $C_{01} = 2.85$, and $D_1 = 7.5 \times 10^{-3}$.

(**a**)

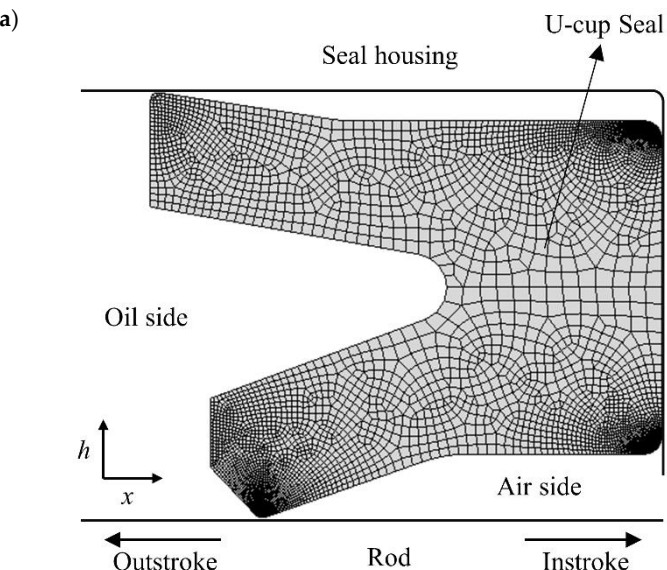

**Figure 1.** *Cont.*

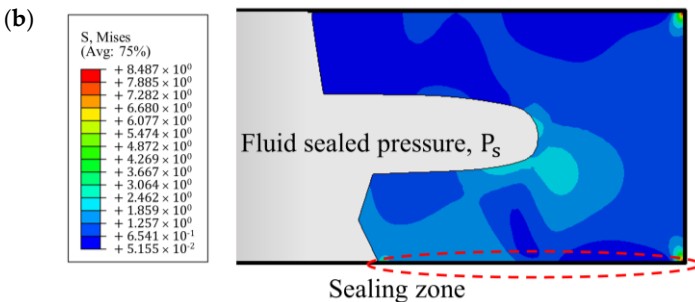

**Figure 1.** Finite element model: (**a**) seal assembly and (**b**) simulation.

Figure 1b shows the von Mises stress after the interference fit simulation analysis and the pressurized process of the rubber seal. It can be seen that stress concentration occurs at the axial end edge in contact with the seal housing. Figure 2 shows the contact pressure in the sealing zone of Figure 1b for sealing pressures of 20 MPa, 25 MPa, 30 MPa, and 35 MPa. The contact force can be calculated by integrating the static contact pressure in Figure 2 and the length of the sealing zone is the length of the section where the contact pressure exists. As the sealing pressure increases, the contact load increases, the seal contracts more, and the length of the sealing zone decreases.

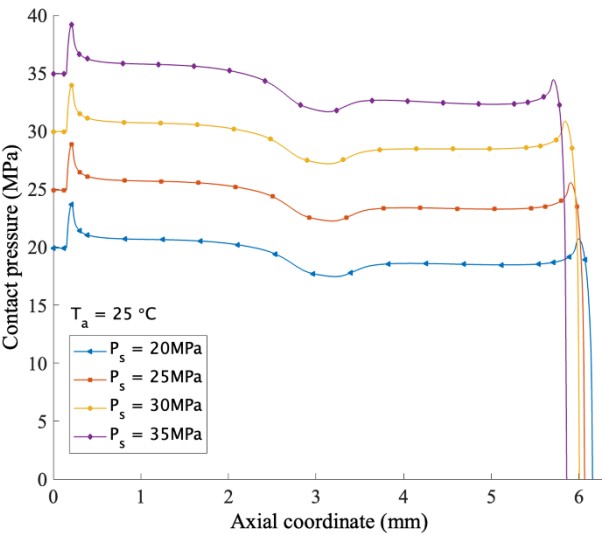

**Figure 2.** Static contact pressure distribution, outstroke.

*2.2. Fluid Mechanics*

2.2.1. Inverse Hydrodynamic Lubrication Method, IHL

The behavior of a fluid in a thin film follows the Reynolds equation. Assuming that the working fluid is an incompressible and Newtonian fluid, the Reynolds equation can be expressed as:

$$\frac{\partial}{\partial x}\left(\frac{h^3}{\mu}\frac{\partial p}{\partial x}\right) + \frac{\partial}{\partial y}\left(\frac{h^3}{\mu}\frac{\partial p}{\partial y}\right) = 6\left(u\frac{\partial h}{\partial x} + 2\frac{\partial h}{\partial t}\right) \tag{2}$$

where $p$ is the pressure at the contact, $h$ is the thickness of the oil film, and $\mu$ is the fluid viscosity. In this study, the Reynolds boundary condition is applied to hydrodynamic lubrication analysis, so it is not calculated below the reference pressure (0 MPa). That is, even if a rapid pressure change occurs during the instroke/outstroke, it is replaced by 0 below the reference pressure. The approach provides an efficiency solution to the oil film pressure, as the Reynolds boundary condition can be easily realized in the numerical solution of the Reynolds equation.

If the seal width is short, a steady state can be assumed and, because it is axisymmetric, Equation (2) can be simplified to Equation (3) [5].

$$\frac{\partial}{\partial x}\left(\frac{h^3}{\mu}\frac{\partial p}{\partial x}\right) = 6u\frac{\partial h}{\partial x} \qquad (3)$$

Integrating Equation (3) gives Equation (4).

$$\frac{dp}{dx} = 6\mu u\frac{h - h_0}{h^3} \qquad (4)$$

where $h_0$ is the oil film thickness at the position where $\partial p/\partial x = 0$, that is, the position where the pressure is the greatest. Substituting Equation (4) into Equation (3) gives:

$$h^3\frac{d^2 p}{dx^2} + 3\frac{dh}{dx}\left(h^2\frac{dp}{dx} - 2\mu u\right) = 0 \qquad (5)$$

The film thickness at the inflection point $I$ is calculated from Equation (6).

$$h_I = \sqrt{2\mu u/\left(\frac{dp}{dx}\right)_{max}} \qquad (6)$$

By substituting Equation (6) into Equation (4), the relationship between $h_0$ and $h_I$ can be obtained.

$$h_0 = \frac{2}{3}h_I = \frac{1}{3}\sqrt{8\mu u/\left(\frac{dp}{dx}\right)_{max}} \qquad (7)$$

Using Equation (4) with nondimensional variables $\lambda$ and $H(x)$, Equation (9) can be obtained to calculate the distribution of the oil film:

$$\lambda = \frac{h_0{}^2}{6\mu u}\frac{dp}{dx}, \quad H(x) = \frac{h(x)}{h_0} \qquad (8)$$

$$\lambda H(x)^3 - H(x) + 1 = 0 \qquad (9)$$

As shown in Equation (9), the Reynolds equation was developed as a cubic algebraic equation for film thickness. The imaginary root must be correctly identified and solved when calculating the root of a cubic polynomial. Otherwise, convergence to the correct solution may not be possible owing to numerical instability. The type set of roots of the cubic polynomial is governed by $\lambda$ in Equation (8), i.e., it depends on the value of the pressure gradient at a specific position $x$. The set of roots can be classified into three types based on the pressure gradient. If the pressure gradient has a positive $x$ value and is smaller than $(\partial p/\partial x)_{max}$, the equation has one negative real root and two positive real roots; therefore, a value that makes the oil film distribution continuous among the two positive real roots. In the section where the pressure gradient has a positive value and is larger than $(\partial p/\partial x)_{max}$, an effective film thickness cannot be defined because the equation has a pair of complex conjugate roots and a negative real root. Therefore, in IHL theory, the oil film thickness in this section is assumed to be the oil film thickness at the inflection point, which corresponds to the horizontal straight section in Figure 3 and the oil film distribution. In the section with a negative pressure gradient, the equation has a pair of complex conjugate roots and one positive real root; thus, a positive real root can be selected while ignoring the complex conjugate number. By processing the root of the third-order polynomial, as in Equation (7), a continuous and smooth oil film distribution can be obtained.

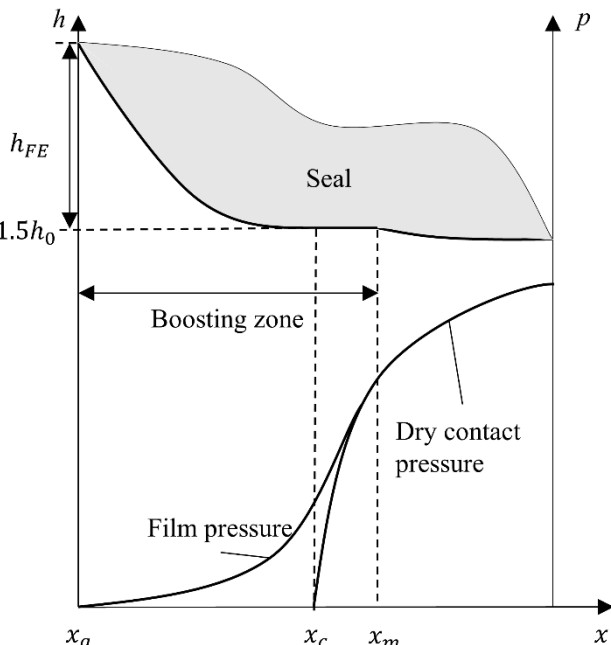

**Figure 3.** Film pressures vs. contact pressures and assumption for the film profile in the boosting zone.

The flow rate can be calculated as follows by integrating the leakage due to the pressure gradient and leakage due to sliding in the circumferential direction.

$$Q = \pi D \left( -\frac{h^3}{12\mu}\frac{\partial p}{\partial x} + u\frac{h}{2} \right) \qquad (10)$$

2.2.2. Boosting Zone

In the IHL analysis, the position of the inflection point and the pressure gradient at that point are important factors that determine the oil film thickness. However, in the FE analysis, the contact pressure at both ends of the rubber seal, the inlet and outlet, decreased very rapidly, making it difficult to describe the smooth pressure change that occurs in fluid lubrication. Accordingly, this problem was solved by introducing a boosting zone in the IHL analysis. Figure 3 shows the FE contact pressure at the inlet and outlet of the contact area and the oil film pressure distribution with the added boosting area. If the reference coordinate system is set at the point where the FE contact pressure reaches a maximum near the region where the fluid enters, the position where the FE contact pressure becomes 0 on the $x$-axis is $x_c$, and the position where the oil film pressure starts is $x_a$. The $x$-axis position $x_I$ of the inflection point exists between $x_c$ and $x_m$; $x_m$ is the point where the pressure $p$ and the pressure gradient $\partial p/\partial x$ in both the FE and IHL analyses coincide. Assuming that $(\partial p/\partial x)_{max}$ at the point where the pressure gradient is continuous, the $x_m$ and the $x_I$ points are at the same position. The oil film distribution in the boosting region can be expressed as follows [9]:

$$h = 1.5h_0 + h_{FE} \ (x_a \leq x \leq x_I) \qquad (11)$$

$$h = 1.5h_0 \ (x_I \leq x \leq x_m) \qquad (12)$$

where $h_{FE}$ is the gap between the seal and rod for a small area just before contact, after the FE static analysis. Since the $h_{FE}$ is directly related to the deformation of rubber seal, it depends on the stroke direction and operating conditions, ambient temperature, and sealed pressure.

### 2.2.3. Greenwood–Williamson Contact Model

The contact force of the asperities in the contact between two rough surfaces can be easily calculated using the Greenwood–Williamson (G–W) model [20]. In the G–W model, the contact between two rough surfaces is equivalent to that of a substantially rough elastic surface and a rigid smooth surface. This rough surface is once again equivalent to the contact of a hard plate with an elastic surface composed of projections with the same radius of curvature $R$. At this time, it is assumed that the height of the asperity follows a Gaussian distribution.

$$\varphi(u) = \frac{1}{\sqrt{2\pi}} e^{-\frac{1}{2}u^2}, u = \frac{z}{\sigma} \tag{13}$$

where $z$ is the height of the rough surface and $\sigma$ is the standard deviation of the asperity height. Let the gap between the two surfaces be $d$, and the dimensionless value be $u_h = d/\sigma$. The force due to the contact of the asperities is calculated as follows, by applying Hertzian contact theory to the statistical equation:

$$P_c = \frac{4}{3} E' \eta R^{\frac{1}{2}} \sigma^{\frac{3}{2}} \int_{u_h}^{\infty} (u - u_h)^{\frac{3}{2}} \varphi(u) du \tag{14}$$

where $E' = (1 - v_1^2)/E_1 + (1 - v_2^2)/E_2$ is the equivalent elastic modulus and $E_1$, $E_2$ and $v_1$, $v_2$ are the elastic modulus and Poisson's ratio of the two objects in relative contact, respectively. Here, $\eta$ is the asperity density, and the projection radius $R$ and projection density $\eta$ can be expressed as a function of $\sigma$ using the dimensionless values $\beta$ and $\gamma$ [21].

$$R = \beta\sigma, \ \ \eta = \frac{\gamma}{\sigma^2} \tag{15}$$

The oil film distribution calculated through lubrication analysis corresponds to the gap $d$ at each position and the bearing force by the asperity contact at each position can be calculated according to Equation (14).

### 2.2.4. Temperature Rise in the Sealing Zone

The friction force between the rubber seal and rod should consider two factors: friction due to fluid viscosity and friction due to asperity contact. Equation (16) shows the shear stress caused by fluid viscosity and shear stress due to surface asperities; the friction force is calculated by integrating them over the stroke length.

$$\tau_v = \frac{h}{2} \frac{\partial p}{\partial x} + \frac{\mu u}{h}, \ \ \tau_c = f p_c \tag{16}$$

$$F = -\pi D l \int_0^l (\tau_v + \tau_c) dx \tag{17}$$

The equation for heat generation rate $q_f$ due to friction generated while the rod reciprocates is as follows:

$$q_f = (\tau_v + \tau_c) u \tag{18}$$

In thermal analysis, it is assumed that all of the heat generated in the seal area is transferred to the semi-infinite solid because generally, the rod has a very high thermal conductivity to the seal [6]. Note that if a rubber seal with high thermal conductivity is used, the generated heat must be calculated by dividing the seal and rod. The heat conduction equation of the seal area for a moving heat source can be expressed as [22]:

$$\begin{aligned} T_{avg} - T_a &= 1.07 \frac{q_f l_c}{k} \left[ \frac{\rho_r c_p u l_c}{k} \right]^{-\frac{1}{2}} \ \ for \ \frac{\rho_r c_p u l_c}{k} > 0.68 \\ &= 0.64 \frac{q_f l_c}{k} ln \left[ \frac{5k}{\rho_r c_p u l_c} \right] \ \ for \ \frac{\rho_r c_p u l_c}{k} < 0.68 \end{aligned} \tag{19}$$

where $T_{avg}$ is the average temperature in the sealing zone, $T_a$ is the ambient temperature, $c_p$ is the specific heat, $\rho_r$ is the density of the rod, $k$ is the thermal conductivity of the rod, and $l_c$ is the half-length of $l$.

Because the fluid viscosity is strongly temperature-dependent, the temperature of the sealing area is used to re-evaluate the fluid viscosity. The viscosity according to temperature was calculated using the Reynolds temperature-viscosity equation [23].

$$\mu = \mu_0 \exp\left(-a\left(T_{avg} - T_0\right)\right) \tag{20}$$

where $\mu_0$ is the viscosity at $T_0$, $\mu$ is the viscosity at $T$, and $a$ is the viscosity–temperature constant.

Figure 4 shows a flowchart of the thermal mixed IHL analysis, and the numerical analysis process is as follows. IHL analysis is performed using input variables, such as static contact pressure, rod movement speed, and operating temperature calculated using the FE software. After introducing the boosting zone concept and calculating the oil film thickness at the inflection point and its location, the oil film distribution for the sealing zone section is obtained according to Equation (9), and by substituting the oil film distribution into Equation (14), a distribution can be obtained. The mixed lubrication effect is reflected by moving the entire oil film distribution up and down in parallel; thus, the sum of the supporting force by the oil film and the asperity contact force is equal to the static contact force obtained through the FE analysis. From the obtained new film distribution and pressure distribution, the viscous shear stress and frictional shear stress of the asperities' contact were calculated and, by substituting them into Equations (17) and (18), the surface friction and heat flux at each position can be determined. The average temperature rise of the sealing zone by heat flux was calculated using Equation (19). By substituting the increased $T_{avg}$ into Equation (20), the updated viscosity in the sealing zone was calculated, and lubrication analysis was performed again with this viscosity. The entire process was repeated until the temperature of the sealing area satisfied thermal equilibrium. As mentioned earlier, this analysis method was assumed to be in a steady state, which is valid when the stroke length is significantly larger than the seal width.

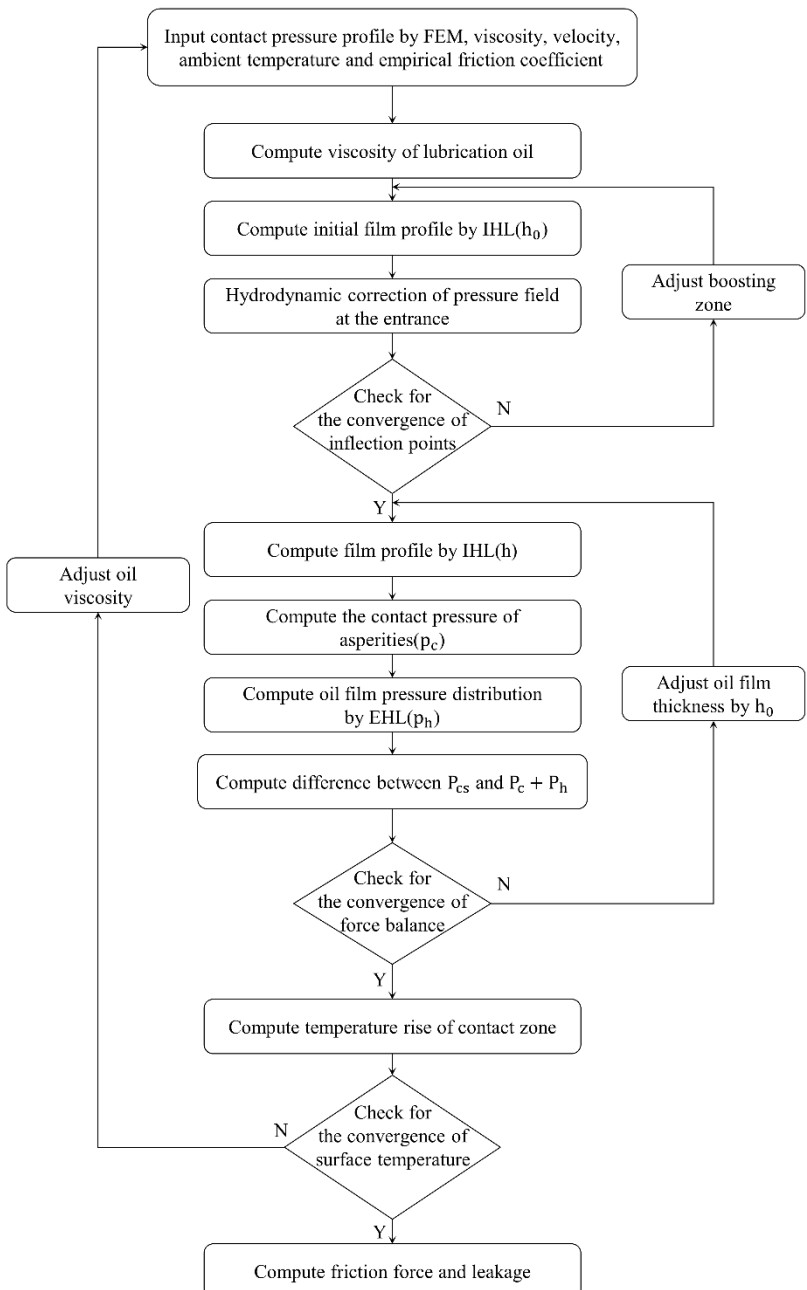

**Figure 4.** Flowchart of thermal mixed IHL.

## 3. Results and Discussion

### 3.1. Analysis Model Description

Analysis of the U-cup seal was performed using the basic parameters listed in Table 1. The rod moved at a constant speed between analyses, and the stroke length was set to 140 mm. The housing and rod were assumed to have a smooth surface and only the rough surface of the seal was considered. Nikas et al. [24]. presented a typical stress–strain diagram of an elastomeric material used for U-cup seals. The rubber material behaves differently depending on external temperature, i.e., it behaves in a significantly different manner at a temperature below zero compared to normal temperature but, at a high temperature above room temperature, the difference in the stress–strain graph is insignificant. In this study, because the analysis was performed at room temperature or higher, it was assumed that the stress–strain relationship did not change depending on the ambient temperature when calculating the static contact pressure through the FE

analysis. However, because the thermal expansion of the seal is considered, other static contact pressure distributions vary with ambient temperature. Through this, numerical analysis results for sealing performance such as flow rate, film thickness, and friction force were presented according to whether the frictional heat effect was considered for different ambient temperatures ($T_a$ = 25 °C, 55 °C, 85 °C, and 115 °C).

**Table 1.** Parameters of the numerical model.

| Parameter and Symbol | Dimensions and Data |
| --- | --- |
| Seal material | HNBR |
| Elastic modulus, $E$ | 43 MPa |
| Poisson's ratio, $\nu$ | 0.499 |
| Rod diameter, $D$ | 25 mm |
| Empirical friction coefficient, $f$ | 0.25 |
| Seal RMS roughness, $\sigma$ | 0.3 µm |
| Sealed fluid | API GL-4 |
| Viscosity, $\mu$ | 0.0771 Pa·s at 25 °C |
| Viscosity at $T_0$, $\mu_0$ | 0.4690 Pa·s at −35 °C |
| Viscosity–temperature constant, $a$ | 0.0301 |
| $\beta$ | 1 |
| $\gamma$ | 1.75 |
| Stroke length, $L$ | 140 mm |
| Sealed pressure, $P_s$ | 20, 25, 30, 35 MPa |
| Rod velocity, $u$ | 0.1, 0.4, 0.7, 1.0 m/s |
| Ambient temperature, $T_a$ | 25, 55, 85, 115 °C |
| Reference temperature, $T_0$ | −35 °C |

*3.2. Effect of Thermal Analysis*

3.2.1. Effect of Rod Velocity and Sealed Pressure

Figure 5 shows the distribution of the oil film at each speed when the seal pressure is 35 MPa and the ambient temperature is 25 °C. As the rod speed increases, a thick film formed owing to improved lubrication performance, and it can be seen that the distribution of the oil film in the outstroke is thicker than that in the instroke; this is because the fluid moves in the direction of seal pressure to atmospheric pressure for the outstroke but moves in the opposite direction for the instroke. Figure 6 shows the amount of leakage according to speed. Here, the reverse pumping rate is the flow rate that occurs from the oil side to the air side and is calculated as the flow rate difference between the outstroke and instroke.

The outstroke flow direction is defined from the oil side to the air side and the Couette flow—caused by the rod surface—and the Poiseuille flow—increased by the pressure gradient at the seal inlet—are in the same direction. The instroke flow direction is defined as the oil side on the air side, opposite to the outstroke flow direction. The Couette flow has a positive value, and the Poiseuille flow has a negative value. In general, because the Couette flow is larger than the Poiseuille flow, the instroke flow has a positive value. This calculation was performed on the assumption that a thin oil film also exists on the air side [18]. It can be observed that the reverse pumping rate increases as the load speed increases. As the load speed increases, the flow rate increases for both instroke and outstroke because a thick film is formed; however, the increase rate is larger because the instroke is larger and thus the reverse pumping rate increases as the load speed increases. An increase in sealed pressure also increases the reverse pumping rate because a greater sealed pressure creates a greater pressure gradient within the contact area. Figure 7 shows

the reverse pumping rate according to sealing pressure and rod velocity; it can be seen that the reverse pumping rate increases as the sealing pressure and rod velocity increase.

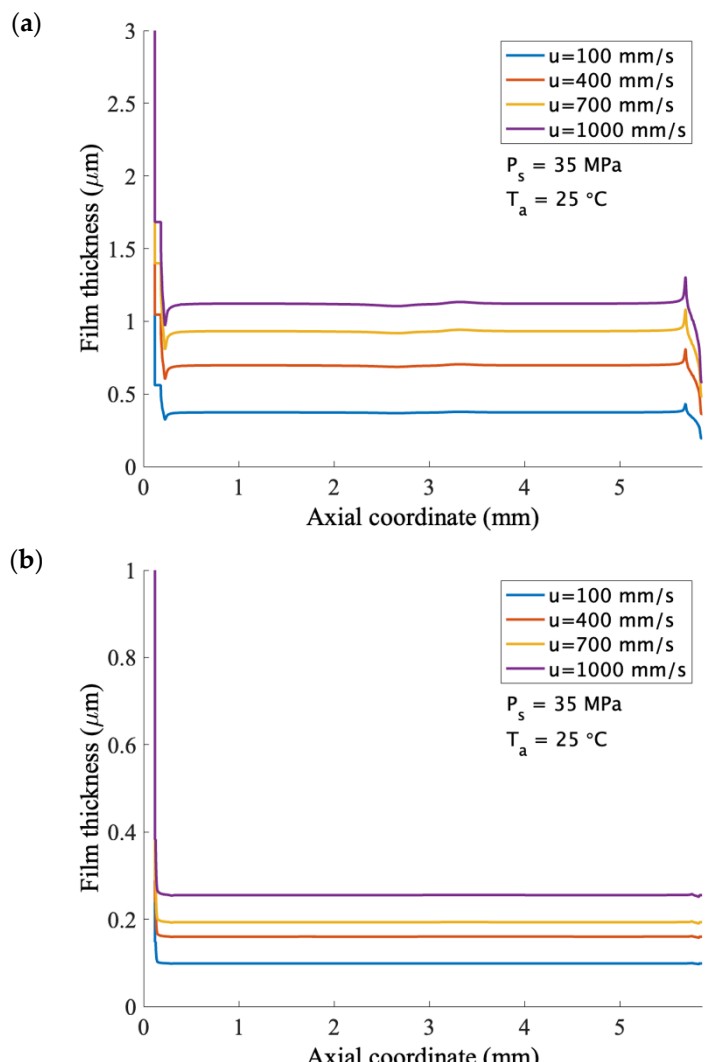

**Figure 5.** Film thickness distribution at different rod velocities (mm/s), 35 MPa, and 25 °C: (**a**) outstroke and (**b**) instroke.

### 3.2.2. Friction Force

In Sections 3.2.2 and 3.2.3, the friction force and leakage according to the ambient temperature were compared when the sealed pressure was 35 MPa and the rod velocity was 1000 mm/s. Figure 8 shows the oil film distribution for the outstroke and instroke, and Figure 9 shows the friction force at the interface according to whether the frictional heat effect is considered for all ambient temperatures. As can be seen from Equation (7), the frictional force in the mixed lubrication state is the sum of the frictional force due to the viscous shear and asperity contact. The film parameter, Λ, obtained by dividing the RMS roughness of the rubber seal at the outstroke of 25 °C; at the ambient temperature by the thickness of the oil film at the center of the contact part, is 3, and a sufficiently thick oil film is formed. At a high temperature of 55 °C or more, the oil film becomes thinner and the frictional force due to the asperity contact increases as the number of asperities in contact increases; this phenomenon becomes more pronounced as the temperature increases. This is because once a thin oil film is formed to the point where asperity contact occurs, and the asperity contact ratio rapidly increases as the thickness of the oil film decreases. In such a situation, the specific gravity of friction due to viscous shear is greatly reduced, and friction

due to asperity contact becomes a dominant factor in the frictional force. In the case of instroke, because the thickness of the oil film is formed with $\Lambda = 1$, even at 25 °C, it can be seen that the asperity contact frictional force dominates at all ambient temperatures, and the total frictional force is much larger than that of the outstroke. Similar to the increase in ambient temperature, the increase in temperature due to frictional heat is also a factor that increases frictional force. Except for the 25 °C outstroke where the thick film is formed, considering the temperature rise of the sealing zone in both the instroke and outstroke, the viscous shear friction force decreases and the asperity contact friction force increases compared to when this effect is neglected. At $\Lambda < 3$, the overall frictional force also tends to increase because the increase in the asperity contact friction force is greater. The higher the ambient temperature, the greater the frictional force increase in consideration of the temperature rise of the sealing zone because the high asperity contact ratio generates a large amount of frictional heat, and the decrease in the thickness of the oil film is large.

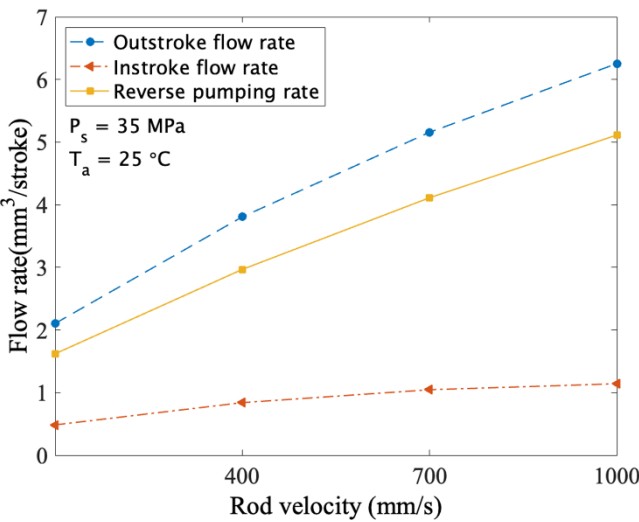

**Figure 6.** Variation of flow rate with rod velocity.

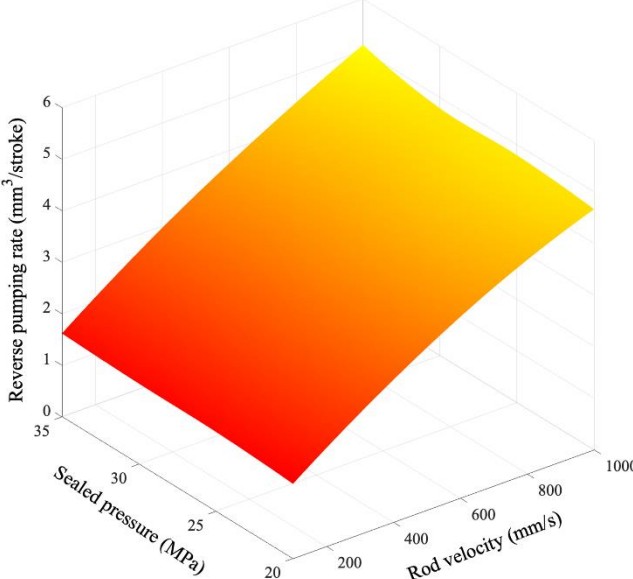

**Figure 7.** Variation of flow rate with sealed pressure and rod velocity.

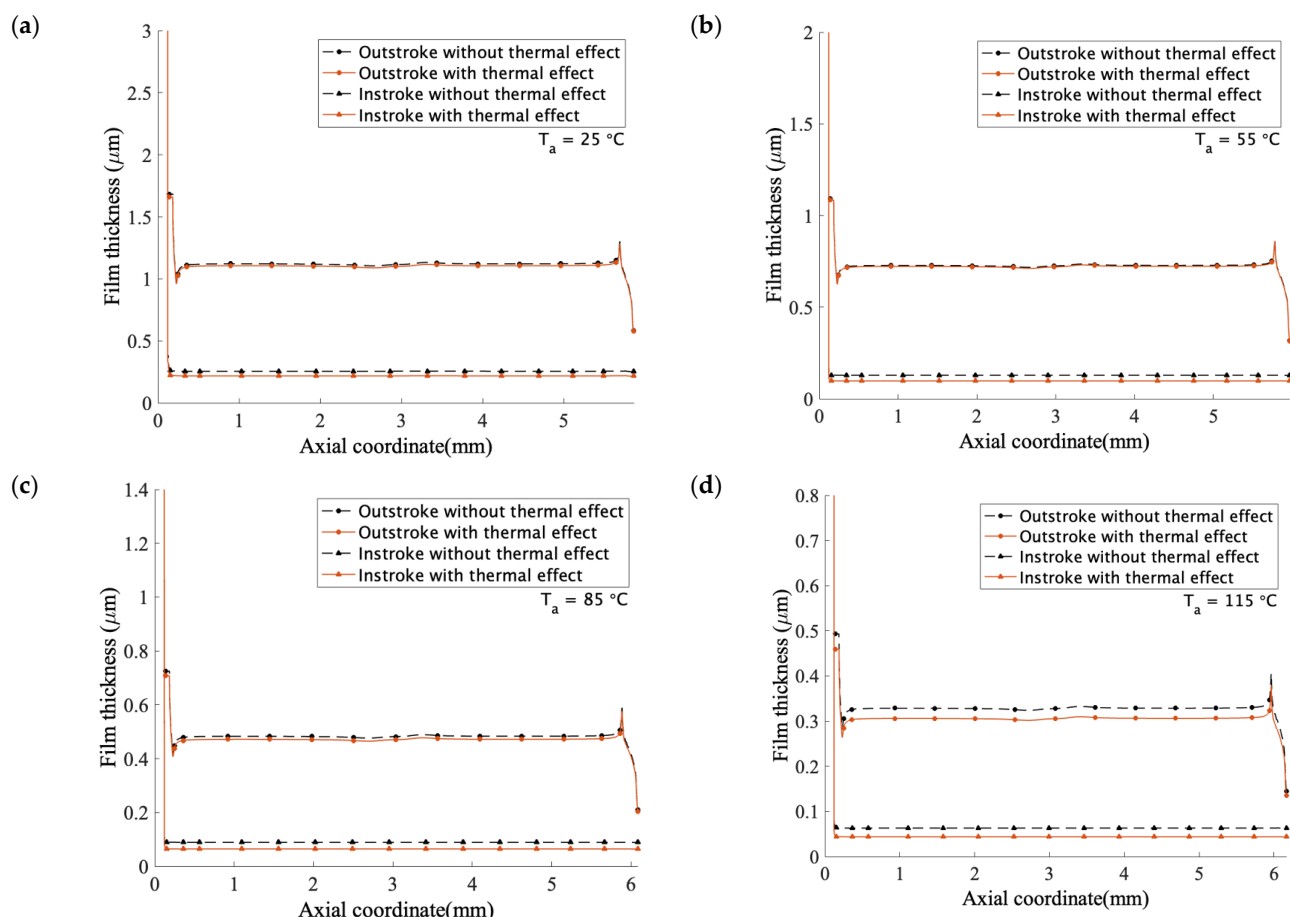

**Figure 8.** Film thickness distribution according to different ambient temperatures and whether thermal effects are considered: (**a**) $T_a = 25$ °C, (**b**) $T_a = 55$ °C, (**c**) $T_a = 85$ °C, and (**d**) $T_a = 115$ °C.

3.2.3. Leakage

Figure 10 shows the difference in flow rate depending on whether the effect of temperature increase due to frictional heat is considered according to the ambient temperature. All flow rates decreased with increasing ambient temperature because the fluid viscosity decreased with increasing temperature and a thinner oil film was formed. Considering the effect of temperature rise, it can be seen that the flow rate decreases in the outstroke; this is because the thickness of the oil film becomes thinner as the temperature of the sealing zone increases, similar to the effect caused by the increase in ambient temperature. In contrast, the flow rate for the instroke increases when the temperature rise is considered. In the outstroke, the leakage caused by Poiseuille and Couette flows is in the same direction. In contrast, in the instroke, the Couette flow introduces the external oil existing on the air side of the outstroke into the pump, but the Poiseuille flow acts in the opposite direction to the Couette flow and causes oil leakage. Considering the effect of temperature rise, the fluid lubrication pressure is reduced by the thinner oil film, which reduces the size of the Poiseuille flow and increases the instroke flow rate. Consequently, the effect of increasing the temperature resulted in reduced leakage in both instroke and outstroke. In the previous section, it was confirmed that the frictional force increased rapidly as the ambient temperature increased. An increase in frictional force means an increase in frictional heat and, thus, the temperature increase at the contact portion increases. Therefore, a higher ambient temperature corresponds to a greater difference in flow rate due to the temperature increase effect. This also suggests that, if the operating conditions are different, the effect of increasing the temperature may be large if the frictional force of the asperity is large, even in a section where the ambient temperature is relatively low. The relative differences in the

reverse pumping rate according to the temperature rise effect were 4.02%, 4.93%, 10.0%, and 20.9% at 25 °C, 55 °C, 85 °C, and 115 °C, respectively.

(**a**)

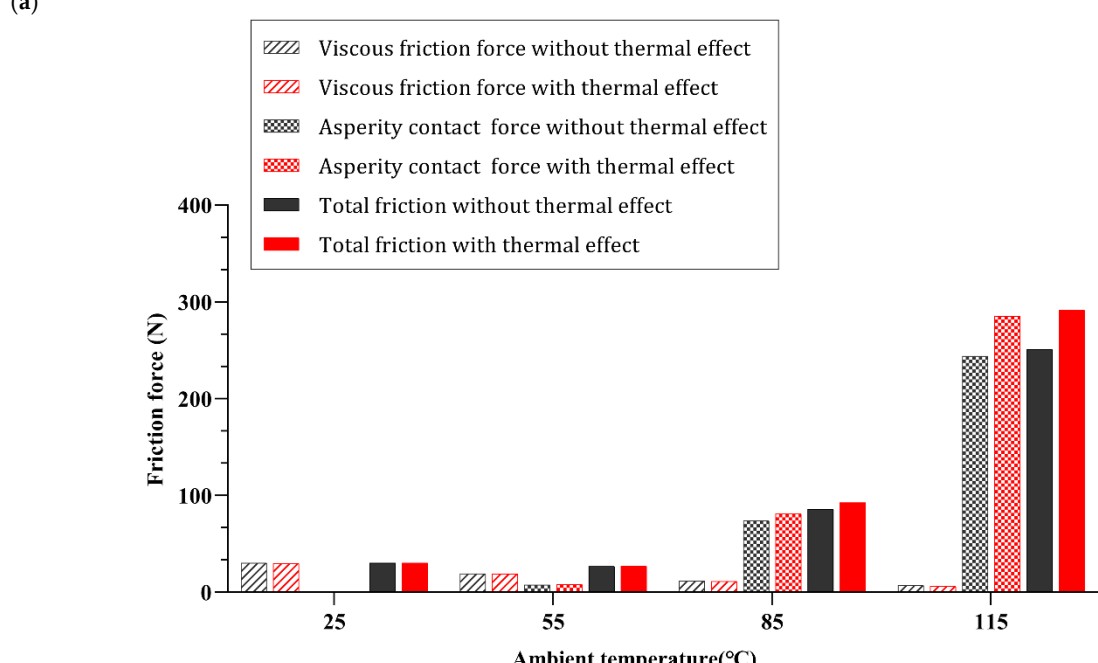

(**b**)

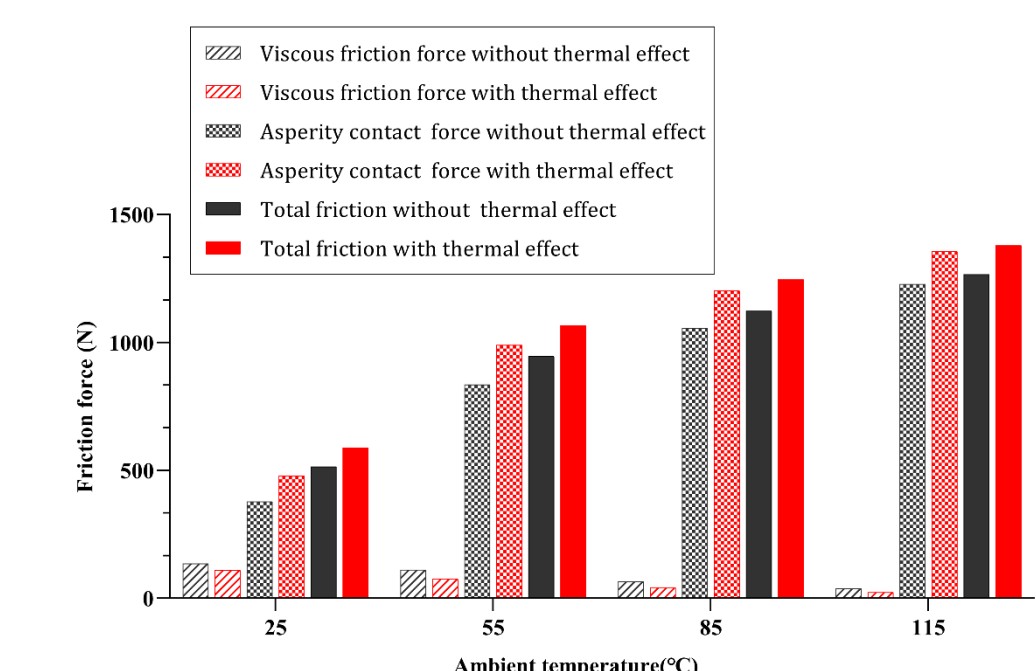

**Figure 9.** Variation of friction force according to different ambient temperatures and whether thermal effects are considered: (**a**) outstroke and (**b**) instroke.

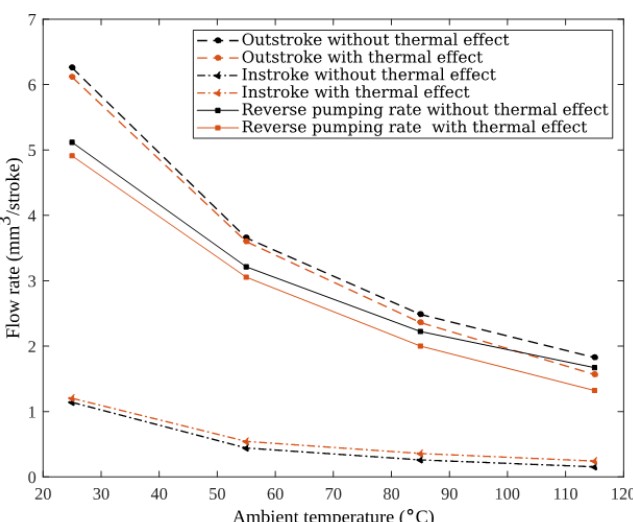

**Figure 10.** Variation in flow rate according to different ambient temperatures and whether thermal effects are considered.

## 4. Conclusions

In this study, a numerical analysis method of thermal mixed IHL considering the effect of temperature rise due to friction in the sealing zone was presented. The sealing performance was analyzed by applying this method to a U-cup within the range of 25–115, sealed pressure of 20–35 MPa, and rod velocity of 0.1–1.0 m/s. From the present results, the following conclusions can be drawn:

(1) The oil film distribution according to operating conditions, such as rod velocity and sealed pressure, was analyzed. Regardless of stroke direction, as the rod velocity and sealed pressure increased, the lubrication performance improved, and a thick film was formed in the sealing zone, thus increasing the reverse pumping rate.

(2) In addition to the rod velocity and sealing pressure, the sealing performance according to the ambient temperature was considered. The ambient temperature plays an important role in the sealing performance. A higher ambient temperature corresponded to the rough surface asperities having a greater effect on the seal owing to the thinner oil film, increasing friction, and reducing the reverse pumping rate.

(3) Considering the effect of temperature rise, it was predicted that the total friction was larger and the reverse pumping rate was smaller; it was further confirmed that the difference increased as the ambient temperature increased. These results show that when predicting the performance of rubber seals, it is very important to consider the thermal effect under the operating conditions of high temperature or partial lubrication with a high asperity contact ratio.

In the above studies, effects—such as thermal expansion of the housing and rod, 3D motion, and transients—were not considered or were deemed negligible. However, and future work will improve the analysis model via a more sophisticated analysis that considers these effects.

**Author Contributions:** Conceptualization, B.K. and Y.Y.; Methodology, B.K., B.L. and Y.Y.; Software, B.K. and B.L.; Validation, B.L.; Investigation, B.K., B.L., Y.C., G.H. and J.P.; Resources, Y.C.; Data curation, J.P.; Writing—original draft, B.K.; Writing—review & editing, B.L., G.H. and Y.Y.; Visualization, G.H., J.P. and Y.Y.; Supervision, J.S. and Y.Y.; Funding acquisition, J.S., Y.C. and Y.Y. All authors have read and agreed to the published version of the manuscript.

**Funding:** This work was supported by the National Research Foundation of Korea (NRF) grant funded by the Korea government (MSIT) (No. 2022R1C1C2003523) and by the Nano·Material Technology Development Program through the National Research Foundation of Korea (NRF) funded by Ministry of Science and ICT (No. 2020M3H4A3106186).

**Institutional Review Board Statement:** Not applicable.

**Informed Consent Statement:** Not applicable.

**Data Availability Statement:** Not applicable.

**Acknowledgments:** The authors express their sincere gratitude to all study participants.

**Conflicts of Interest:** The authors declare no conflict of interest.

## Nomenclature

| | |
|---|---|
| $W$ | general form of the strain–energy function for incompressible materials |
| $I_1$, $I_2$ | first and second deviatoric strain invariants |
| $J$ | elastic volume ratio |
| $C_{10}$, $C_{01}$, $D_1$ | material coefficients of the Mooney–Rivlin model |
| $p$ | hydrodynamic pressure |
| $p_c$ | asperity contact pressure |
| $x$ | axial direction |
| $y$ | circumferential direction |
| $h$ | film thickness |
| $h_0$ | film thickness at the position where $\partial p/\partial x = 0$ |
| $Q$ | flow rate |
| $u$ | sliding rod velocity |
| $D$ | diameter of the rod |
| $E$ | elastic modulus |
| $\nu$ | Poisson's ratio |
| $\eta$ | asperity density |
| $R$ | asperity radius |
| $\sigma$ | root-mean-square (RMS) roughness of sealing element surface |
| $\beta$ | dimensionless asperity radius |
| $\gamma$ | dimensionless asperity density |
| $\tau_c$ | asperity contact shear stress |
| $\tau_v$ | viscous shear stress |
| $f$ | friction coefficient |
| $F$ | friction force |
| $q_f$ | heat generation rate due to friction |
| $T_{avg}$ | average temperature in the sealing zone |
| $T_a$ | ambient temperature |
| $c_p$ | specific heat |
| $\rho_r$ | density of the rod |
| $k$ | thermal conductivity of the rod |
| $l$ | sealing length |
| $l_c$ | half-length of $l$ |
| $\mu$ | lubricant dynamic viscosity |
| $a$ | viscosity–temperature constant |

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
