# Peer review of "Numerical Analysis via Mixed Inverse Hydrodynamic Lubrication Theory of Reciprocating Rubber Seal Considering the Friction Thermal Effect"

_applsci, doi:10.3390/app13010153_

Round 1

Reviewer 1 Report

The current work investigates the performance of U-cup rubber seals using the Inverse Fluid Lubrication theory, to estimate film thickness based on pressure distribution, and the Greenwood-Williamson contact model, to account for surface roughness on estimation of friction forces. Film distribution, leakage and friction force are estimated and the effect of sealed pressure, rod velocity and ambient temperature on these parameteres is examined. The necessity to account for thermal effects is mainly due to reduction of the oil viscosity and the consequent reduction of fluid film load carrying capacity, as higher ambient temperatures correspondend to thinner oil films, which increased friction due to asperity contact. Overall, paper is well organized. Nonetheless, some points should be reviewed and discussed.

The following points should be elucidated. They are organized according to its location on the manuscript.

1. The first sentence of the abstract (page 1, line 9) reads: "This study investigates how the shape and physical properties of typical U-cup seals [...] affect sealing performance." Perhaps this sentence should be rewritten, as it is misleading. The analysis and results presented in the paper are for a U-cup seal with the given geometric and physical properties shown in Table 1 of the manuscript. An analysis on the effect of the shape of the seal (or variation of its geometric dimensions) or the effect of the rubber physical parameters (or variation of its elastic modulus) is not conducted in the manuscript. Also, only a U-cup is considered. Other types of seals are also not considered. The abstract should emphasize the novelty of the work and which parameters are being investigated to evaluate a U-cup seal performance.

2. Page 2, lines 51-52 reads: 'the oil film gradient or squeeze [...] is the main pressure-generating mechanism'. Notice that pressure is not a thermodynamic property that can be 'generated' In sliding bearings, such quoted mechanisms increase the lubricant pressure. Please, correct  it (change pressure generation to pressure increase). Review it throughout the manuscript.

3. Page 3, Eq 1: please define after the equation what W is. 

4. Page 2, line 60: 'FE analysis'. Every other acronym has been defined on the manuscript. This is the first time the FE acroynm appears, but it is not defined.

5. On Fig. 1a, indicate the system of coordinates (x y z directions) on the figure.

6. Was variation of viscosity with pressure considered on the analysis, or only temperature (Eq. 20, page 7)?

7. In the flowchart of Figure 4 and the text preceding it, the pressure profile is estimated using the commercial software ABAQUS, because "the elastic deformation of the rubber surface is much larger than the fluid film thickness, it is hardly affected by the presence of a thin oil film". Therefore, FEA analysis gives the pressure profile. Was the rest of the analysis done on ABAQUS or was this numerical analysis conducted on another software? Please, clarify it.

8. Please, include in Table 1 the oil type.

9. Check Figures 9a,b. On Figure 9a, include a space between 'with thermal' on the second entry of the legend. On Figure 9b, 'viscous friction force without thermal effect' appears twice and 'total friction with thermal effect' also appears twice. Finally, on both figures, should it be more appropriate to change the x-label of Figures 9a,b to 'Ambient Temperature'?

10. Is cavitation of the oil film possible to occur during the instroke/outstroke? If so, how was it considered? Can such effect change the rubber seal geometry or the results presented, affect either the lubricant pressure or temperature? Please, discuss.

11. Manuscript does not present a model validation, the correctness of the theoretical model can not be verified. Perhaps it should contain a section or an appendix with experimental validation?

12. Please, include a table of nomenclature, with the symbols used in the equations and the acronyms defined in the text.

Author Response

Point 1: The first sentence of the abstract (page 1, line 9) reads: "This study investigates how the shape and physical properties of typical U-cup seals [...] affect sealing performance." Perhaps this sentence should be rewritten, as it is misleading. The analysis and results presented in the paper are for a U-cup seal with the given geometric and physical properties shown in Table 1 of the manuscript. An analysis on the effect of the shape of the seal (or variation of its geometric dimensions) or the effect of the rubber physical parameters (or variation of its elastic modulus) is not conducted in the manuscript. Also, only a U-cup is considered. Other types of seals are also not considered. The abstract should emphasize the novelty of the work and which parameters are being investigated to evaluate a U-cup

Response 1: Manuscript revised. (line 17)

Point 2: Page 2, lines 51-52 reads: 'the oil film gradient or squeeze [...] is the main pressure-generating mechanism'. Notice that pressure is not a thermodynamic property that can be 'generated' In sliding bearings, such quoted mechanisms increase the lubricant pressure. Please, correct it (change pressure generation to pressure increase). Review it throughout the manuscript.

Response 2: Manuscript revised. (line 61)

Point 3: Page 3, Eq 1: please define after the equation what W is. 

Response 3: Manuscript revised. (line 124)

Point 4: Page 2, line 60: 'FE analysis'. Every other acronym has been defined on the manuscript. This is the first time the FE acroynm appears, but it is not defined.

Response 4: Manuscript revised. (line 70)

Point 5: On Fig. 1a, indicate the system of coordinates (x y z directions) on the figure.

Response 5: Manuscript revised. (Fig1.(a))

Point 6: Was variation of viscosity with pressure considered on the analysis, or only temperature (Eq. 20, page 7)?

Response 6: In this study, viscosity was only considered as a function of temperature.

Point 7: In the flowchart of Figure 4 and the text preceding it, the pressure profile is estimated using the commercial software ABAQUS, because "the elastic deformation of the rubber surface is much larger than the fluid film thickness, it is hardly affected by the presence of a thin oil film". Therefore, FEA analysis gives the pressure profile. Was the rest of the analysis done on ABAQUS or was this numerical analysis conducted on another software? Please, clarify it.

Response 7: In IHL, it is assumed that the contact pressure obtained through FE analysis is equal to the hydrodynamic pressure calculated as the film thickness. The process of deriving the film thickness with a given contact pressure through ABAQUS, and the process of load convergence and surface temperature convergence considering the asperity contact force are performed with MATLAB in-house code.

Point 8: Please, include in Table 1 the oil type.

Response 8: Manuscript revised. (Table 1)

Point 9: Check Figures 9a,b. On Figure 9a, include a space between 'with thermal' on the second entry of the legend. On Figure 9b, 'viscous friction force without thermal effect' appears twice and 'total friction with thermal effect' also appears twice. Finally, on both figures, should it be more appropriate to change the x-label of Figures 9a,b to 'Ambient Temperature'?

Response 9: Manuscript revised. (Fig. 9)

Point 10: Is cavitation of the oil film possible to occur during the instroke/outstroke? If so, how was it considered? Can such effect change the rubber seal geometry or the results presented, affect either the lubricant pressure or temperature? Please, discuss.

Response 10: In this study, the Reynolds boundary condition is applied to lubrication analysis, so it is not calculated below the reference pressure (0 MPa). That is, even if a rapid pressure change occurs during the instroke/outstroke, it is replaced by 0 below the reference pressure. Through this, cavitation can be confirmed in the film thickness, but it is difficult to check the change in seal shape or cavitation wear.

Point 11: Manuscript does not present a model validation, the correctness of the theoretical model can not be verified. Perhaps it should contain a section or an appendix with experimental validation?

Response 11: We used the previously verified IHL theory, and this method has been verified several times in the papers (examples) cited in our paper, so we did not add a separate verification part. In addition, the main purpose of our study was to figure out the tendency by numerical investigation based on the IHL theory, and additional research is needed for verification with experiments.

Point 12: Please, include a table of nomenclature, with the symbols used in the equations and the acronyms defined in the text.

Response 12: Manuscript revised. (line 467)

Reviewer 2 Report

The authors have conducted a numerical analysis based on the mixed inverse hydrodynamic lubrication (IHL) theory to study the influence of the shape and physical properties of typical U-cup seals and operating conditions on the sealing performance. The paper can be accepted for publication after revision by addressing the following comments.

(1) The “IHL” in the title should be replaced by “inverse hydrodynamic lubrication”.

(2) The influence of mesh on the numerical simulation results should be done and an optimized mesh conditions should be selected. However, the authors have neglected this part.

(3) The authors have emphasized that their model considers the thermal effect during contact, however, they have not presented the corresponding results in the paper. At least, the temperature evolution is required to be provided. What is the maximum temperature would reach? Does the temperature affect the physical properties of lubricant, U-cup seal and rod steel?

(4) Honestly speaking, according to the results shown in Figure8 and Figure 9, there is no significant difference in the film thickness, asperity contact and total friction between the numerical analysis with and without consideration of thermal effect. Thus, what is the critical contribution of this paper? Please explain the advantage of the model used in this paper.

(5) The format of references needs to be improved. Their publication years should be also provided.

Author Response

Point 1: The “IHL” in the title should be replaced by “inverse hydrodynamic lubrication”.

Response 1:  Manuscript revised. (line 2)

Point 2: The influence of mesh on the numerical simulation results should be done and an optimized mesh conditions should be selected. However, the authors have neglected this part.

Response 2: The mesh convergence was performed when further improvements did not appreciably improve the accuracy of the calculations. The contact pressure change was observed by increasing the mesh density of the edge of the seal where contact with the housing or rod occurs, and the calculation speed was improved by reducing the mesh density in the center with less deformation. Therefore, the mesh density (a total of 6966 mesh elements) was selected for simulation in the present work.

Point 3: The authors have emphasized that their model considers the thermal effect during contact, however, they have not presented the corresponding results in the paper. At least, the temperature evolution is required to be provided. What is the maximum temperature would reach? Does the temperature affect the physical properties of lubricant, U-cup seal and rod steel?

Response 3:  The effect of ambient temperature on fixed sealed pressure and rod velocity was confirmed in Section 3.2. The maximum temperature change reached for each ambient temperature of 25 to 115 °C were 12, 22, 25, and 28 °C, and the temperature change affects the lubrication viscosity. The change in viscosity due to temperature rise was the largest at the ambient temperature of 115 ℃, up to 57% compared to the initial condition, and accordingly, the change in frictional force and flow rate according to whether or not the thermal effect was considered was confirmed to be the largest.

In this study, the thermal expansion of the U-cup seal according to the initial ambient temperature was considered, although the temperature rise did not affect the physical properties of the U-cup seal and surrounding structures.

Point 4:  Honestly speaking, according to the results shown in Figure8 and Figure 9, there is no significant difference in the film thickness, asperity contact and total friction between the numerical analysis with and without consideration of thermal effect. Thus, what is the critical contribution of this paper? Please explain the advantage of the model used in this paper.

Response 4: As the effect of temperature rise was considered, the total friction was predicted to be larger and the reverse pumping rate to be smaller, and it was confirmed that the difference increased as the ambient temperature increased. The higher the ambient temperature, the thinner the oil film increases the impact of the asperities on the seal's rough surface, resulting in greater frictional forces and reduced reverse pumping rate. These results show that when predicting the performance of rubber seals, it is very important to consider thermal effects under operating conditions of partial lubrication, such as high temperatures or high asperity contact ratios.

Point 5: The format of references needs to be improved. Their publication years should be also provided.

Response 5:  Manuscript revised. (Reference)

Reviewer 3 Report

The presented subject is up-to-date. The authors discussed a thermal effect on friction of reciprocating rubber seal. They also presented interesting results of conducted analyses. However, some issues require explanation and more extensive discussion.

Some expressions used in the paper are unclear. There are parts of the paper that are hard to read.

The following issues require broader explanation:

line 181:

“where ℎ?? is the gap between the seal and rod for a small area just before contact, after the FE static analysis”

 "small area just before contact" - This was not explained clearly enough. How does the area change depending on assumed conditions?

Figure 1. Finite element model: (a) seal assembly and (b) simulation.

The arrows placed in the Figure, showing direction of the movement of outstroke and instroke can be misleading.  

Figure 2. Static contact pressure distribution …

On the x axis the values were given in mm. There is lack of values in the Figure 1 which depict the cross-section of a seal.

How can pressure distribution be related to the width of a seal cross-section?

Table 1. - there is lack of sufficient explanation concerning the value of empirical friction coefficient, shown in the table. How was it measured?

line 307:

“The film parameter, Λ, obtained by dividing the RMS roughness of the  rubber seal at the outstroke of 25℃ at the ambient temperature by the thickness of the oil  film at the center of the contact part, is 3, .... “

Would it not be better to additionally present graphically the discussed interactions?

line 329:

Figure 8. Film thickness distribution according to different ambient temperatures

The differences between film thickness for particular cases are hard to read from the Figures. Why were the differences not discussed in the same way as they were discussed for pumping rate later on in the text?

The presented conclusions are obvious and largely known. It would be better to indicate the novelty of this research.

The text contains a lot of minor errors, which still make reading more difficult. For example, line 416

Ranters, A. F. C.  -  This is Kanters who is the author of the publication.

Author Response

Point 1: line 181:

“where ℎ?? is the gap between the seal and rod for a small area just before contact, after the FE static analysis”

 "small area just before contact" - This was not explained clearly enough. How does the area change depending on assumed conditions?

Response 1: As mentioned in the manuscript, ℎ?? is the gap between seal and rod for a small area immediately before contact calculated by FE static analysis. This is the area where the seal and rod do not come into contact, and it depends on operating conditions such as sealing pressure and ambient temperature.

Point 2: Figure 1. Finite element model: (a) seal assembly and (b) simulation.

The arrows placed in the Figure, showing direction of the movement of outstroke and instroke can be misleading. 

Response 2: The arrows in the figure indicate the moving direction of the reciprocating rod. In Figure 1a, the coordinate system of the figure (x, h direction) has been added.

Point 3: Figure 2. Static contact pressure distribution …

On the x axis the values were given in mm. There is lack of values in the Figure 1 which depict the cross-section of a seal.

How can pressure distribution be related to the width of a seal cross-section?

Response 3: Figure 2 shows the contact pressure in the sealing zone of Figure 1(b) for sealing pressures of 20 MPa, 25 MPa, 30 MPa and 35 MPa. Therefore, the length of the sealing zone is about 6 mm, and it gets longer as the sealing pressure increases.

Point 4: Table 1. - there is lack of sufficient explanation concerning the value of empirical friction coefficient, shown in the table. How was it measured?

Response 4: It was not directly measured due to laboratory conditions. It was referenced from the paper of R.F. Salant mentioned in the introduction.

(Ref. : Yang, B.and Salant, R. F., Elastohydrodynamic lubrication simulation of O-ring and U-cup hydraulic seals,Engineering Tribology, 2011)

Point 5: line 307:

“The film parameter, Λ, obtained by dividing the RMS roughness of the  rubber seal at the outstroke of 25℃ at the ambient temperature by the thickness of the oil  film at the center of the contact part, is 3, .... “

Would it not be better to additionally present graphically the discussed interactions?

Response 5: The use of the film parameter Λ to describe the film thickness according to the ambient temperature is for quantitative comparison in the text. Fig. 8 shows the film thickness for ambient temperature, so that comparison by condition is possible. Therefore, it is considered unnecessary to add film parameters as a graph.

Point 6: line 329:

Figure 8. Film thickness distribution according to different ambient temperatures

The differences between film thickness for particular cases are hard to read from the Figures. Why were the differences not discussed in the same way as they were discussed for pumping rate later on in the text?

Response 6: Fig. 8 shows the distribution of the film thickness according to the change in ambient temperature for a fixed rod velocity, and Fig. 9 and Fig. 10 show the friction force and flow rate considering this film thickness. I think that the differences between film thickness for particular cases has been sufficiently discussed by comparing friction and flow rate.

Point 7: The presented conclusions are obvious and largely known. It would be better to indicate the novelty of this research.

Response 7: The novelty of this study is the investigation of the effect of the mixed IHL method, considering the temperature increase due to frictional heat in the sealing zone. It is a well-known result in lubrication theory, but it is emphasized that the IHL method combined with FE analysis reflects the change in viscosity due to heat generation by considering the effect of the rough surface.

Point 8: The text contains a lot of minor errors, which still make reading more difficult. For example, line 416

Ranters, A. F. C.  -  This is Kanters who is the author of the publication.

Response 8: Manuscript revised. (References)

Round 2

Reviewer 1 Report

The revised manuscript can still be improved.

Figure 4 should be improved. The text inside the diamond shapes in Figure 4 is crossing the shape boundaries. Increase the shape size.

In Figure 10, the legend should not be above the lines in the plot.

Authors' response stated that the Reynolds boundary conditions were used to model cavitation on the film. This is not stated anywhere in the manuscript.

Oil type is not defined in Table 1, nor the values parameters used to define the viscosity-temperature relation.

What are the values of the parameters mu_0, a and T0 in Eq. 20, used in the simulations?

Perhaps the authors could present a graph with the contact pressure change and computing time with mesh density. Or, at least, state in the manuscript the mesh convergence analysis was conducted to choose the 6966 mesh elements used in the simulations.

What is the rod velocity and seal pressure used in the simulations for the results presented in Sections 3.2.1 (Friction force) and 3.2.2 (Leakage)? 

Author Response

Point1: Figure 4 should be improved. The text inside the diamond shapes in Figure 4 is crossing the shape boundaries. Increase the shape size.

Response1: Figure 4 has been modified to reflect your comment.

Point2: In Figure 10, the legend should not be above the lines in the plot.

Response2: Figure 10 has been modified to reflect your comment.

Point3: Authors' response stated that the Reynolds boundary conditions were used to model cavitation on the film. This is not stated anywhere in the manuscript.

Response3: Manuscript revised. (line 147.)

For the efficiency analysis, it is added to the manuscript that Reynolds boundary condition is applied in hydrodynamic lubrication analysis.

Point4: Oil type is not defined in Table 1, nor the values parameters used to define the viscosity-temperature relation.

What are the values of the parameters mu_0, a and T0 in Eq. 20, used in the simulations?

Response4: Manuscript revised. (Table 1.)

This oil is used in this simulation for API GL-4 and viscosity-temperature relation is added to Table 1. The parameters used in Eq. 20 are shown in Table 1.

Point5: Perhaps the authors could present a graph with the contact pressure change and computing time with mesh density. Or, at least, state in the manuscript the mesh convergence analysis was conducted to choose the 6966 mesh elements used in the simulations.

Response5: Manuscript revised. (line 117) 

Accepting your opinion, it is stated in the manuscript that a mesh convergence analysis was performed to select 6966 mesh elements used in the simulation.

Point6: What is the rod velocity and seal pressure used in the simulations for the results presented in Sections 3.2.1 (Friction force) and 3.2.2 (Leakage)? 

Response6: Manuscript revised. (line 323) 

Operating conditions not listed in the text have been modified. Also, there was an error in the section number and it was corrected.

Reviewer 3 Report

The responses given by the Authors partially explain their point of view. However, more extensive discussion of the issues mentioned in the text of the paper can make it easier for a reader to follow the conducted analyses. 

That is why, the issues indicated previously should be more broadly discussed in the text of the paper. 

Response 1: As mentioned in the manuscript, ℎ is the gap between seal and rod for a small area immediately before contact calculated by FE static analysis. This is the area where the seal and rod do not come into contact, and it depends on operating conditions such as sealing pressure and ambient temperature.

This should be presented more broadly in the text. 

All the explanations presented by the Authors should also be clearly discussed in the text of the publication. 

Author Response

Point1: As mentioned in the manuscript, ℎ is the gap between seal and rod for a small area immediately before contact calculated by FE static analysis. This is the area where the seal and rod do not come into contact, and it depends on operating conditions such as sealing pressure and ambient temperature.

This should be presented more broadly in the text. 

All the explanations presented by the Authors should also be clearly discussed in the text of the publication. 

Response1: The manuscript has been revised by adding the following sentences (line 203) .

“Since the  is directly related to the deformation of rubber seal, it depends on the stroke direction and operating conditions, ambient temperature and sealed pressure.”

Round 3

Reviewer 1 Report

everything seems correct now.